# An Image Segmentation Method Based on Improved Regularized Level Set Model

**Lin Sun [1,2,*]** , **Xinchao Meng [1]** , **Jiucheng Xu [1,2]** and **Shiguang Zhang [1,2]**

1   College of Computer and Information Engineering, Henan Normal University, Xinxiang 453007, Henan, China; 18749607976@163.com (X.M.); jiuchxu@gmail.com (J.X.); sg201106@163.com (S.Z.)
2   Engineering Technology Research Center for Computing Intelligence and Data Mining, Xinxiang 453007, Henan, China
*   Correspondence: linsunok@gmail.com or sunlin@htu.edu.cn; Tel.: +86-373-332-6190

**Abstract:** When the level set algorithm is used to segment an image, the level set function must be initialized periodically to ensure that it remains a signed distance function (SDF). To avoid this defect, an improved regularized level set method-based image segmentation approach is presented. First, a new potential function is defined and introduced to reconstruct a new distance regularization term to solve this issue of periodically initializing the level set function. Second, by combining the distance regularization term with the internal and external energy terms, a new energy functional is developed. Then, the process of the new energy functional evolution is derived by using the calculus of variations and the steepest descent approach, and a partial differential equation is designed. Finally, an improved regularized level set-based image segmentation (IRLS-IS) method is proposed. Numerical experimental results demonstrate that the IRLS-IS method is not only effective and robust to segment noise and intensity-inhomogeneous images but can also analyze complex medical images well.

**Keywords:** image segmentation; level set; distance regularization term; energy functional

## 1. Introduction

Image segmentation plays an important role in image processing and image understanding [1]. It is an important portion of almost all computer vision fields for real-world engineering applications, ranging from object extraction to complex medical images, satellite images, video and traffic surveillance systems, etc. [2]. As a preprocessing stage, it segments an image into different homogeneous regions according to a certain consistency. At present, it is still a challenge to deal with noise, low contrast with weak edges, intensity inhomogeneity, and complex backgrounds for image segmentation [3]. Over the past several decades, many great techniques and methodologies for image segmentation have been developed to overcome these drawbacks.

The recently developed methods of image segmentation can be roughly divided into two categories: traditional methods and soft computing methods [2]. Since the former methods are simple and easy to implement, they have been widely applied and provide exact solutions to potential and practical applications. In general, considering their form of operation, these methods can be grouped into different classes of clustering, thresholding, boundary tracking, region-based segmentation, and edge-based segmentation, among others [4]. In the last few years, many clustering-based models have been used in image segmentation. In particular, those algorithms based on K-means clustering, fuzzy c-means clustering (FCM) and spectral clustering are the most widely used in image segmentation problems [5]. Wang et al. [6] presented an adaptive segmentation method for crop disease images based on K-means clustering to improve the accuracy and stability of disease spot

segmentation. As the FCM algorithm is sensitive to noise and selection to the initial cluster centers, some improvements to the FCM model have been investigated for both the estimation of the intensity inhomogeneity and segmentation of magnetic resonance image data [7]. Ramudu et al. [8] combined kernel FCM clustering with particle swarm optimization to develop a global region-based image segmentation method with a fast level set model. Hu et al. [9] proposed an adaptive kernel-based FCM clustering with spatial constraints model to automatically control the influence of the neighborhood pixels on the central pixel. However, its drawbacks include the slow convergence of the iteration and over smoothing when suppressing noise [10]. He et al. [11] investigated an incremental spectral clustering method for stream image segmentation. It is known that for traditional spectral clustering, a scaling parameter needs to be fixed artificially, and obtaining its optimal value is very difficult in a Gaussian kernel function. Thus, handling the scaling parameter is a sensitive task. To solve this issue, Zelnik-Manor and Perona [12] constructed a self-tuning method for spectral clustering and image segmentation with a local scaling parameter. Goyal et al. [13] studied a fuzzy similarity measure-based spectral clustering method for noisy image segmentation, which improves the robustness to the influence of noise. However, most of these models share some common drawbacks, such as the cluster number needing to be given in advance and the sensitivity to selecting initial cluster centers. In addition, when low-level features are considered to make the representations in most algorithms, some crucial information may be lost. As a result, the segmentation performance may be degraded for some images to some extent. Over several years, the thresholding-based image segmentation method has drawn broad attention, and numerous thresholding techniques have been developed. Since they are simple and robust to noisy images, the threshold-based segmentation methods have been widely applied. The most representative method is the Otsu between-class variance. However, its computation grows exponentially when more threshold values are incorporated [14]. Tobias and Seara [15] presented image segmentation by histogram thresholding using fuzzy sets according to the similarity between gray levels, which is easy to implement and has a low computational burden. However, it does not consider the spatial contextual information in the threshold selection process. Singla and Patra [16] designed a fast context-sensitive threshold selection technique to solve image segmentation problems. Ananthi et al. [17] investigated *L*-interval-valued intuitionistic fuzzy sets and set the least entropy as the threshold to segment the image. However, the method may lead to increased computation time, especially when dealing with multithreshold image segmentation [18]. Gao et al. [19] introduced an enhanced artificial bee colony optimizer into multilevel threshold image segmentation. For the traditional interactive image segmentation, the robots only work for region-based methods, excluding the important class of approaches that rely on the boundary tracking paradigm. Then, Spina and Falcao [20] proposed robot users to simulate human user behavior when segmenting an image through the addition of anchor points close to the object's boundary. Miranda et al. [21] developed an optimum user-steered boundary tracking approach for image segmentation, which simulates the behavior of water flowing through a riverbed. However, the inefficiency of these approaches lies in the fact that they cannot deal with real-life complex problems that are tolerant of partial truths, imprecision, uncertainty, and approximations [2]. Note that none of these methods are good at segmenting a variety of images, and they usually perform well in one class of images while performing poorly in another class of images. In recent years, the active contour model (ACM) [22] has become one of the most promising frameworks and effective and accurate methods for image segmentation. Currently, most existing ACMs can be categorized into two basic classes: edge-based models and region-based models, according to image features for segmentation.

The basic idea of the ACM framework is to attract the curve toward the true boundary of an object based on an energy minimization model. The two main shortcomings of ACM algorithms are as follows: (1) the sensitivity to the initial position and (2) the difficulties related to topological changes [23]. Until now, scholars [22–41] have investigated a large amount of ACM algorithms for the performance improvement of image segmentation. In 1988, Osher and Sethian [24] first presented a level set model, which is the most successful and important part of the ACM family. The evolution of

the level set function is controlled by partial differential equations that describe the image, which drives the zero level set to move toward the target edge of the image. This model is a numerical method for interface tracking and shape modeling that avoids tracking and parameterizing the curve as it evolves and can flexibly deal with topological changes. Hence, the calculation accuracy is high, and the related algorithm is stable. In addition, the numerical calculations required by the level set method can be performed on a Cartesian grid without the need to parameterize the points on the contour. Due to the abovementioned advantages, it has become a very useful tool for segmenting images [25–28]. To solve all these issues, it is necessary to explore new ways or directions for image segmentation to achieve better overall performance. Thus, we focus on studying the ACM algorithm in this paper to identify a new image segmentation method.

Thus far, there are many variations to the level set model, which can roughly be divided into three categories: edge-based methods [29–31], region-based methods [32–35], and hybrid methods [36–44]. The geodesic active contour [45] is the most typical of the edge-based methods, but these methods have one very marked disadvantage: they are very sensitive to the initial position of the level set and can easily divulge the weak edges [46]. To address the above defects, the Chan Vese (CV) model [33], as the most typical of region-based level set methods, was studied. The CV method is insensitive to the initial position and independent of the image gradients so that it can handle images with weak edges [46]. The hybrid level set methods associate the edge-based models with region-based models for image segmentation [41–44].

Note that this CV method, as a global segmentation model, has all the merits of the region-based level set model, but this model cannot segment heterogeneous images well when the intensity is not homogenous [23]. Many real-world images, especially medical images, exhibit intensity inhomogeneity. Researchers [47–55] have proposed different solutions to address this issue. For example, Lv et al. [47] integrated fuzzy decisions and the local energy functional and presented a robust ACM to segment preprocessed vessel images. Vese and Chan [48] introduced the Mumford and Shah method to develop an image segmentation algorithm with a multiphase level set model. However, this algorithm is still time-consuming to compute, which limits its application. Li et al. [49] studied a region-based ACM and designed a local binary fitting (LBF) method, which enables the extraction of accurate local image information. However, the model is sensitive to the initial position and noisy images. Zhang et al. [50] introduced a local image fitting (LIF) energy functional to extract the local image information, and proposed a Gaussian filtering method for variational level set to regularize the level set function, which can be interpreted as a constraint on the differences between the original image and the fitting image. Zhang et al. [51] exploited a local image region statistics-based improved ACM (LSACM) and provided a level set method in the presence of intensity inhomogeneity that is robust to noise while suppressing the intensity overlapping to some extent. He et al. [52] investigated an improved scheme for a region-scalable fitting method. However, a defect of this method is its sensitivity to the initialization position. Shi and Pan [53] investigated a local and global binary fitting model that can segment an inhomogeneous image with less iteration and avoid the reinitialization of the curve evolution. Li et al. [27] designed a type of level set evolution called distance regularized level set evolution (DRLSE), which eliminates the costly process of reinitialization, uses the simple difference method to decrease the computational complexity, and improves the evolution efficiency [54]. According to this analysis, most of the existing approaches are sensitive to the initialization. Furthermore, some methods are unable to handle images with noise and intensity inhomogeneity. All of these drawbacks limit their practical applications. Thus, we focus on solving the abovementioned issues in this paper.

In this study, an image segmentation method based on an improved regularized level set model is investigated. A type of energy functional in our model consists of three parts: an external energy term, an internal energy term and a distance regularization term. Since the external energy is introduced, the inhomogeneous images can be efficiently segmented. Moreover, the reinitialization step can be avoided by employing a distance regularization term with a new potential function. As a result,

the time costs and iterations are greatly reduced. Notably, a curve of the level set evolution can automatically stop on the true boundaries of objects. The results of our study indicate that the proposed IRLS-IS method not only segments inhomogeneous images and noisy images but also effectively analyzes complex medical images.

The rest of the article is organized as follows. In Section 2, we briefly review some background knowledge. A regularized level set model is improved to avoid the reinitialization step. The proposed IRLS-IS model maintains a stable evolution speed during the evolution process and successfully completes image segmentation. All of these processes are investigated in Section 3. Several experiments are conducted on some synthetic and real images, and the results are discussed in Section 4. Section 5 presents our conclusions.

## 2. Related Work

### 2.1. Chan Vese (CV) Model

The original CV model is on the basis of the evolution of the curve. A continuous curve is used to represent the segmentation target area, and it is controlled by the energy functional. Then, the energy functional of the curve's independent variable transforms the segmentation problem into a function for minimizing the energy. The image will be divided when the energy functional is close to the minimum [33]. The CV model is an ACM based on global information. By adding the area and the length of the curve, the CV model can be used as a smooth constraint on the evolution of the curve for detecting objects.

Assume that $u$ is a given image, and then the energy functional [40,55] can be written as

$$
\begin{aligned}
E^{CV}(u,C) = \quad & \mu \int_\Omega \delta(\varphi)|\nabla(\varphi)|dxdy + v \int_\Omega H(\varphi)dxdy + \lambda_1 \int_\Omega |u - c_1|^2 H(\varphi)dxdy \\
& + \lambda_2 \int_\Omega |u - c_2|^2 (1 - H(\varphi))dxdy,
\end{aligned}
\tag{1}
$$

where $\mu$, $v$, $\lambda_1$ and $\lambda_2$ denote the corresponding coefficients, all of which are positive constants. Generally, $\lambda_1 = \lambda_2 = 1$, $\Omega$ is a given image domain, $\phi$ is a level set function, $H$ is the Heaviside function, $\delta$ is the Dirac delta function, $c_1$ describes the average gray value in the target area and the background area in evolution curve $C$, and $c_2$ denotes the average gray value in the target area and the background area outside evolution curve $C$. For the energy functional, the first two terms are regular terms that represent the length of the curve and the area inside the curve, respectively. The latter two terms are collectively called the fidelity terms and are responsible for attracting the evolution curve $C$ to the target contour.

According to the calculus of variations and the gradient descent model, the partial differential formula can be obtained by minimizing $E^{cv}(u, C)$. Then, the partial differential equation is expressed as

$$
\frac{\partial \varphi}{\partial t} = \delta(\varphi)\left[\mu \cdot div\frac{\nabla \varphi}{|\nabla \varphi|} - \lambda_1 |u - c_1|^2 + \lambda_2 |u - c_2|^2\right],
\tag{2}
$$

where $div\frac{\nabla \varphi}{|\nabla \varphi|}$ is the curvature of the level set, and $c_1$ and $c_2$ need to be updated at each iteration.

To ensure the stability and the validity of the level set evolution, $\phi$ must be near the SDF, especially close to the zero level set model. The SDF has good properties, such as $|\nabla \varphi| = 1$, which make the rate of change of the level set function uniform everywhere. Therefore, the SDF is used to keep the evolution of the level set model stable [56,57]; that is, the initial level set function is an SDF. After several iterations, the level set function is periodically reinitialized to the SDF. However, this reinitialization forces the CV model to be an SDF. As Gomes and Faugeras [58] revealed a disagreement between the theory and its implementation in the practical application of the level set method, a general scheme to solve these issues so far has not been obtained, and the process of the reinitialization is usually performed in a special manner. It is therefore necessary to investigate a new distance regularization term to avoid reinitialization [18].

## 2.2. Distance Regularization Term

Arnold [59] notes that the function φ, which meets $|\nabla\varphi| = 1$, is an SDF plus a constant [60]. According to the above analysis, Li et al. [61] designed an algorithm without reinitialization. The algorithm uses the following energy penalty term to weaken the deviation between the level set function and the SDF [62,63]. Meanwhile, the introduction of the penalty term can improve the speed of the image segmentation. Then, the energy penalty term is given as

$$P(\varphi) = \int_\Omega p(|\nabla\varphi|)dxdy, \tag{3}$$

where the potential function is described as

$$p(s) = \frac{1}{2}(s-1)^2. \tag{4}$$

Thus, with the calculus of variations and the steepest descent model [64], a gradient flow formula corresponding to the energy functional $P(\phi)$ is obtained as

$$\frac{\partial\varphi}{\partial t} = div(d_p(|\nabla\varphi|)\nabla\varphi), \tag{5}$$

where $div(\bullet)$ is the divergence operator and $d_p$ is a function defined by Li et al. [27,54] according to the function relationship, written as

$$d_p(s) = \frac{p'(s)}{s}. \tag{6}$$

From Equations (4) and (6), $d_p(s) = 1 - \frac{1}{s}$ can be obtained as the diffusion ratio of the level set evolution.

## 3. Proposed Method

### 3.1. New Distance Regularization Term

In this subsection, this deviation between the level set function and the SDF has been offset by the regularization of the traditional level set model; therefore, compared with the traditional level set function, our method as an extended CV model has the advantages of a more accurate computation and a more stable evolution process.

It is known that for the level set model, a potential function must have a point with a minimum on $s = 1$, and the goal of constructing a new potential function is to keep $|\nabla\varphi| = 1$ in the vicinity of the zero level set model. Then, owing to a new potential function, the regularized level set function can reserve an SDF throughout the process of evolution. Thus, a new potential function is constructed as

$$p(s) = \frac{1}{2}(s-3)^2 + 6\ln(s+2) - (2 + 6\ln 3). \tag{7}$$

Due to the existing proportional relationship between the diffusion ratio and the potential function, a diffusion ratio can be defined as

$$d_p(s) = 1 - \frac{3}{s+2}. \tag{8}$$

Based on the energy penalty term Equation (3) and the defined potential function Equation (8), a new distance regularization term is defined as

$$P(\varphi) = \int_\Omega (\frac{1}{2}(|\nabla\varphi| - 3)^2 + 6\ln(|\nabla\varphi| + 2) - (2 + 6\ln 3))dxdy. \tag{9}$$

The new distance regularization term corrects the deviation of the level set function and the SDF. Meanwhile, this term ensures that the level set evolves without requiring periodic initialization and

can avoid a sharp or flat shape. Therefore, the reinitialization step can be avoided during the level set evolution.

### 3.2. New Energy Functional

The distance regularization term is employed to ensure the stability and the validity of the level set evolution [65]. To accurately segment an image, the energy functional should contain an external energy term and force the evolution curve to move toward the target edge.

On the basis of the distance regularization term, by combining the internal and external energy terms, a new energy functional is given as

$$E(\varphi) = \mu P(\varphi) + \lambda E_{\text{int}}(\varphi) + \nu E_{ext}(\varphi). \tag{10}$$

To minimize the impact of noise in the process of segmenting images, the external energy term $E_{ext}$ ($\phi$) is utilized, where the Laplacian of a Gaussian (LoG) filter is contained. The LoG filter is employed to maintain the sensitivity during the evolution of the level set and then drive the zero level set to move toward the target edge. Based on the above descriptions, assume that $\phi$ is a level set function, and then the $E_{ext}$ ($\phi$) is expressed as

$$E_{ext}(\varphi) = \int_{\Omega} (\Delta G_{\sigma} * I)(H(-\varphi)) dx dy, \tag{11}$$

where $\Delta$ is the Laplacian operator, $G_{\sigma}$ denotes the Gaussian kernel function, $G_{\sigma}*I$ describes the convolution operation of $I$ with $G_{\sigma}$, and the Heaviside function $H(x)$ can be described as

$$H(x) = \frac{1}{2} \left[ 1 + \frac{2}{\pi} \arctan \frac{x}{\varepsilon} \right]. \tag{12}$$

Then, the internal energy term is expressed as

$$E_{\text{int}}(\varphi) = \int_{\Omega} \delta(\varphi) |\nabla \varphi| dx dy, \tag{13}$$

where $|\nabla \varphi|$ is the gradient mode of $\phi$, and $\delta(x)$ is the Dirac delta function and can be written as

$$\delta(x) = \frac{1}{\pi} \frac{\varepsilon}{\varepsilon^2 + x^2}. \tag{14}$$

Thus, a new energy functional is constructed by combining our proposed distance regularization term with the internal and external energy terms. The new energy functional is described as

$$\begin{aligned} E(\varphi) = \quad & \mu \int_{\Omega} (\frac{1}{2}(|\nabla(\varphi)| - 3)^2 + 6\ln(|\nabla(\varphi)| + 2) - (2 + 6\ln 3)) dx dy \\ & + \lambda \int_{\Omega} \delta(\varphi) |\nabla(\varphi)| dx dy + \nu \int_{\Omega} (\Delta G_{\sigma} * I)(H(-\varphi)) dx dy, \end{aligned} \tag{15}$$

where $\mu$, $\lambda$ and $\nu$ are greater than zero.

The calculus of variations and the steepest descent model [64] are introduced to minimize $E(\phi)$, and the corresponding partial differential equation is expressed as

$$\frac{\partial \varphi}{\partial t} = \mu div((1 - \frac{3}{(|\nabla \varphi| + 2)}) \nabla \varphi) + \lambda \delta(\varphi) div(\frac{\nabla \varphi}{|\nabla(\varphi)|}) + \nu \delta(\varphi)(\Delta G_{\sigma} * I). \tag{16}$$

### 3.3. Regularized Level Set Model-Based Image Segmentation Algorithm

The flowchart of the proposed IRLS-IS method is outlined in Figure 1.

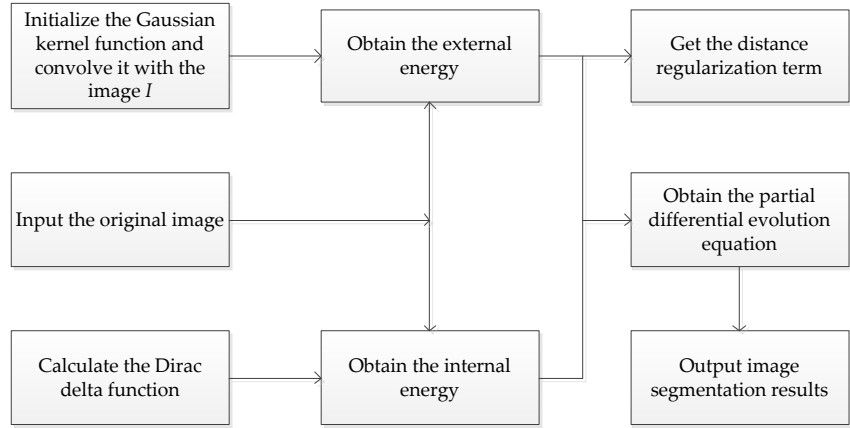

**Figure 1.**　　The process of the proposed improved regularized level set-based image segmentation method.

After the abovementioned procedures of image segmentation, the detailed steps of the proposed IRLS-IS method can be summarized as follows.

---

**Algorithm 1. IRLS-IS**

---

**Input:** An original image
**Output:** The result of image segmentation
Step 1: Initialize parameters $\delta$, $\mu$, $\lambda$ and $\nu$
Step 2: Set the level set function $\phi_0(x, y) = 1$
Step 3: ComCompute the Dirac delta functionStn
Step 4: For n = 1: iterNum
Step 5: Calculate $\Delta G_\sigma * I$, that is, the Gaussian kernel function in [66] is convolved with image $I$, and then the Laplacian operator is applied
Step 6: Compute $div \frac{\nabla \varphi}{|\nabla \varphi|}$ to determine the curvature of the level set function
Step 7: Compute $d_p(s)$ using Equation (8)
Step 8: Update the level set function
Step 9: If min $\frac{\partial \varphi}{\partial t}$ is found by using Equation (16), then output the result
Step 10: Else return to Step 4
Step 11: End for

---

For the IRLS-IS algorithm, the computational complexity mainly focuses on Step 5. In Algorithm 1, Step 5 is the most time-consuming to calculate the Gaussian convolution, and its time complexity is approximately $O(K^2 \times N)$, where typically $K < 7$ when a $K \times K$ Gaussian convolution kernel is implemented in the spatial domain on an image with $N$ pixels. Hence, the time complexity of IRLS-IS is close to $O(N)$. Note that the computational complexity of reinitialization is $O(N^2)$ [50]. Since the IRLS-IS has eliminated the reinitialization step, its time costs and iteration operations are drastically reduced. Therefore, the computational complexity of our IRLS-IS method is lower than that of the other related level set models [33,48,67–69].

## 4. Experimental Results

### 4.1. Experiment Preparation

To demonstrate the performance of our IRLS-IS model on various representative synthetic and real images of different characteristics, more comprehensive results for all contrasted algorithms should be obtained and analyzed. Following the experimental techniques for image segmentation designed by Ji et al. [34], these selected images are mostly corrupted with one or more degenerative characteristics, including additive noise, low contrast, a low signal-to-noise ratio, weak edges,

and intensity inhomogeneity. Some related parameters that are set in the process of image segmentation are illustrated in Figure 2; the original images can be found in [3,42,50]. All of the compared models in this paper are performed in MATLAB version R2014a in a Windows 7 environment using a 3.20 GHz Intel (R) CPU with 4 GB of RAM.

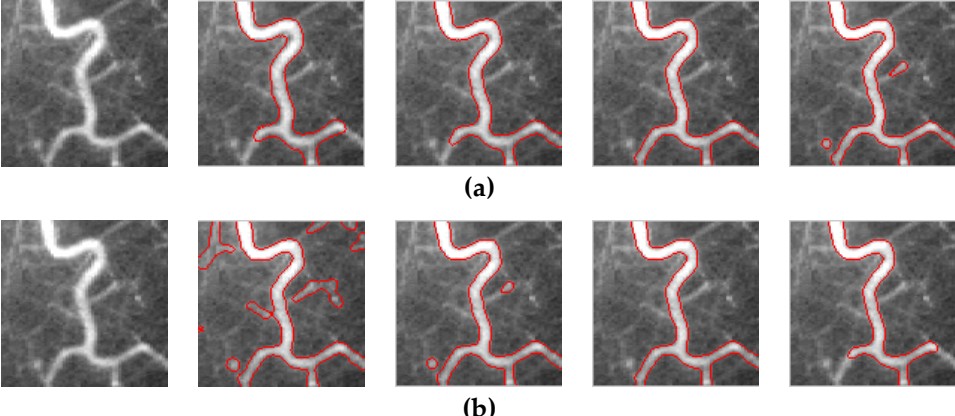

**Figure 2.** The segmentation results of a blood vessel using different parameter values: (**a**) The original image (left) and the segmentation results of our method with $v = 0.15$ and different $\sigma$; (**b**) The original image (left) and the segmentation results of our method with $\sigma = 2.6$ and different $v$.

Figure 2 shows the IRLS-IS method involves two parameters to be manually set. Figure 2a illustrates the original image (left) and the experimental results (right) by setting $\sigma = 4$, 3.2, 2.6, and 2 with $v = 0.15$. Figure 2b shows the original image (left) and the experimental results (right) by setting $v = 0.4$, 0.2, 0.15, and 0.1 with $\sigma = 2.6$. The other related parameters are set as $\Delta t = 5$, $\lambda = 5$ and $\mu = 0.01$.

It is well known that the objective of image segmentation is to segment the whole image domain into several distinct regions in line with the regional consistency. To the best of our knowledge, most existing image segmentation techniques [2] only segment the target regions of images, which are usually not identified by tags and have no special contents. Our proposed IRLS-IS method is different from the approaches for the semantic segmentation of an image [70–73], in which the task is to predict the pixel-level category labels and recognize the objects in the image and segment them.

### 4.2. Segmentation of Single-Objective Images

In this subsection, the goal is to test the effectiveness of our method in terms of the segmentation of the single-objective images. The DRLSE model [27] can maintain the regularity of the level set function, particularly the desirable signed distance property in the vicinity of the zero level set. The LBF model [49], as a region-based ACM for image segmentation in a variational level set framework, effectively uses local image information to segment the intensity-inhomogeneous images. The LIF model [50] has less computational complexity than the LBF model and does not need reinitialization for image segmentation. The LSACM [51] includes two-phase level set models and yields closed-form solutions for the estimated parameters in the distribution. These representatives ACM algorithms are four state-of-the-art level set methods published recently for image segmentation. They show improvements over the classical ACM and are specially selected based on the level set method for comparison experiments. The chosen parameters for the four models can be found in [27,49–51]. The experimental results are demonstrated in Figure 3, and the original images shown in Figure 3a can be found in [52,74]. Following the experimental techniques for image segmentation in [26–51] in which the boundaries of objects were labeled in red, we used the same techniques to mark the boundaries of the target regions in this paper.

As shown in Figure 3, the Row 3, and Row 4 images are two real-world images, and the other images are synthetic images. Some unnecessary objects are obtained by the DRLSE and the LBF models in the second and third columns, and the corresponding experimental results are illustrated in

Figure 3b,c, while the LIF model cannot deal with the second and third images in the fourth column of Figure 3. The reason for the segmentation failure is that the LIF model relies on global information, and so it cannot segment images with a weak boundary. The segmentation results are ideal for the fourth image in the fifth column, and the experimental results are demonstrated in Figure 3e. As seen from Figure 3f, the IRLS-IS model succeeds in segmenting the images. It follows that perfect segmentation is primarily due to the external energy term, which can efficiently solve this defect of the weak boundary and obtain good segmentation results. This outcome states that the IRLS-IS algorithm efficiently completes the segmentation of single-objective images with a weak boundary.

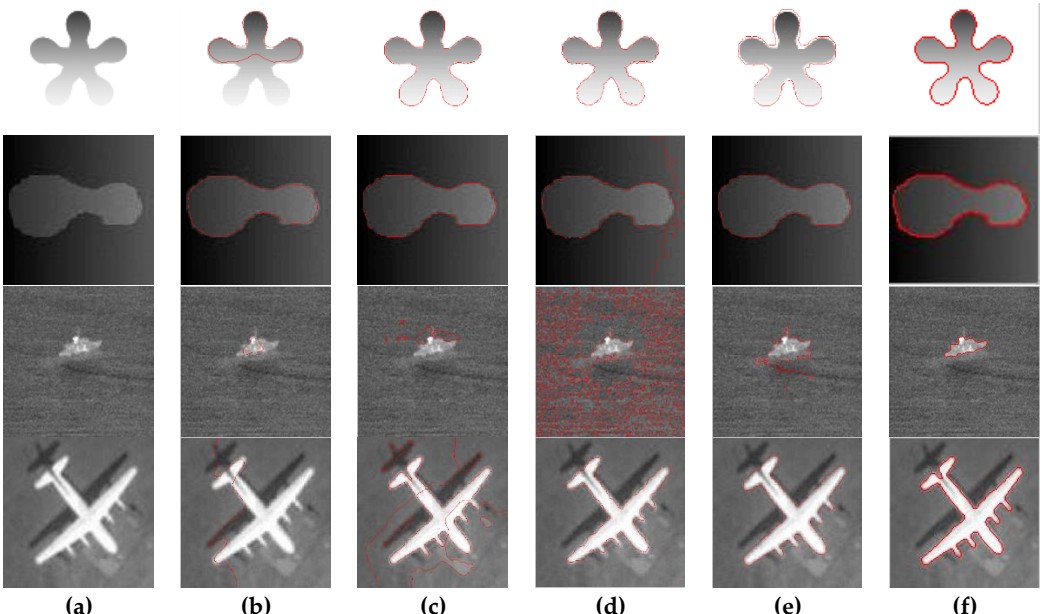

  **(a)**   **(b)**   **(c)**   **(d)**   **(e)**   **(f)**

**Figure 3.** Comparisons of the five models in segmenting single-objective images: (**a**) The original image; (**b**) The segmentation results of the DRLSE model; (**c**) segmentation results of the LBF model; (**d**) The segmentation results of the LIF model; (**e**) The segmentation results of the LSACM model; (**f**) The segmentation results of the IRLS-IS model. DRLSE: distance regularized level set evolution; LBF: local binary fitting; LIF: local image fitting; LSACM: local image region statistics-based improved active contour model; IRLS-IS: improved regularized level set-based image segmentation.

### 4.3. Segmentation of Multiobjective Images

This portion of our experiments considers the segmentation of multiobjective images. The IRLS-IS method is compared with the abovementioned four methods (DRLSE, LBF, LIF, and LSACM). The original multiobjective images and the segmentation results of the five models are demonstrated in Figure 4, and the original images shown in Figure 4a are from [28,34]. The images are synthetic images. As shown in Figure 4b,d, the DRLSE and LIF models failed to segment the multiobjective images. Row 3 of Figure 4c shows that the LBF model cannot detect the true boundary. The object boundaries are precisely extracted by the LSACM algorithm and our IRLS-IS model, and the results are illustrated in Figure 4e,f. The results indicate that our IRLS-IS method effectively extracts the objects in the multiobjective images.

### 4.4. Segmentation of Noisy Images

The following subsection describes an experiment on the segmentation of noisy images. We still choose the above four models in Subsection 4.3 for some comparison experiments. Figure 5 shows the original images with different intensity noise and the comparison results of the five segmentation methods, where the original images without noise in Figure 5a are coming from [75,76]. In Figure 5, Row 1 and Row 6 are the original images and the segmentation results, respectively. Row 2 and Row 3

present noisy images (zero mean, variance 0.04 and 0.06) and the segmentation results, respectively. Row 4 and Row 5 also present noisy images (zero mean, variance 0.1 and 0.3) and the segmentation results. Figure 5b,e show that the DRLSE and LSACM algorithms cannot eliminate the interference of noise and fail to segment the images. It can be observed from Figure 5c,d that the LBF and LIF models could analyze the images without Gaussian noise well, but for the Gaussian noisy images, they perform segmentation poorly. As shown in Figure 5f, the object boundaries are accurately extracted by our IRLS-IS method. The improvements in the results are mainly due to the external energy term, which can successfully address the noise problem. It can be clearly seen that the IRLS-IS model effectively eliminates the noise interference and completes the segmentation of the noisy image.

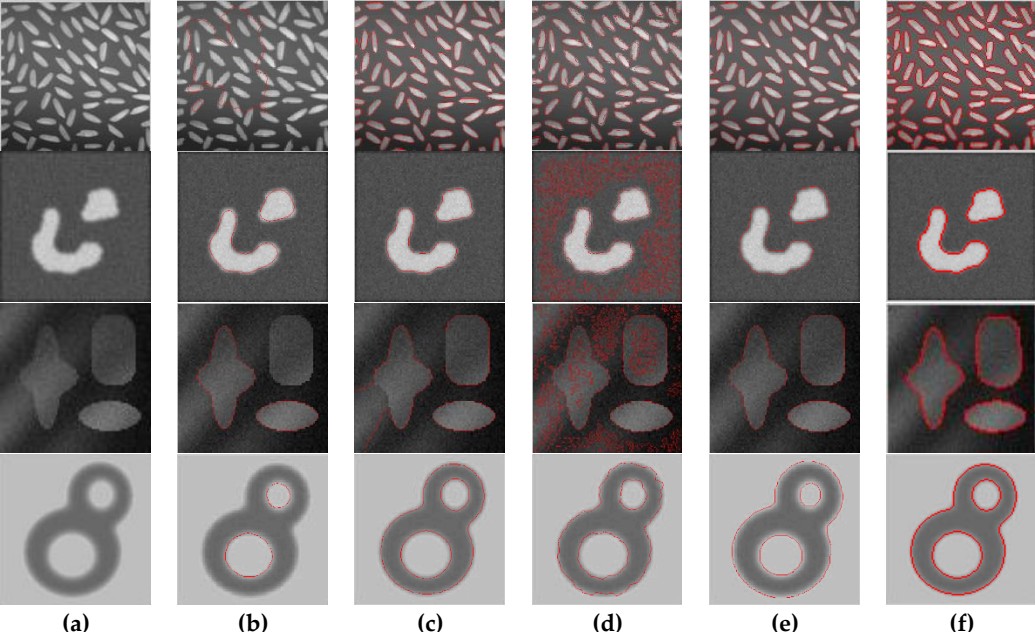

| **(a)** | **(b)** | **(c)** | **(d)** | **(e)** | **(f)** |

**Figure 4.** Comparisons of the five models in segmenting multiobjective images: (**a**) The original image; (**b**) The segmentation results of the DRLSE model; (**c**) The segmentation results of the LBF model; (**d**) The segmentation results of the LIF model; (**e**) The segmentation results of the LSACM model; (**f**) The segmentation results of the IRLS-IS model.

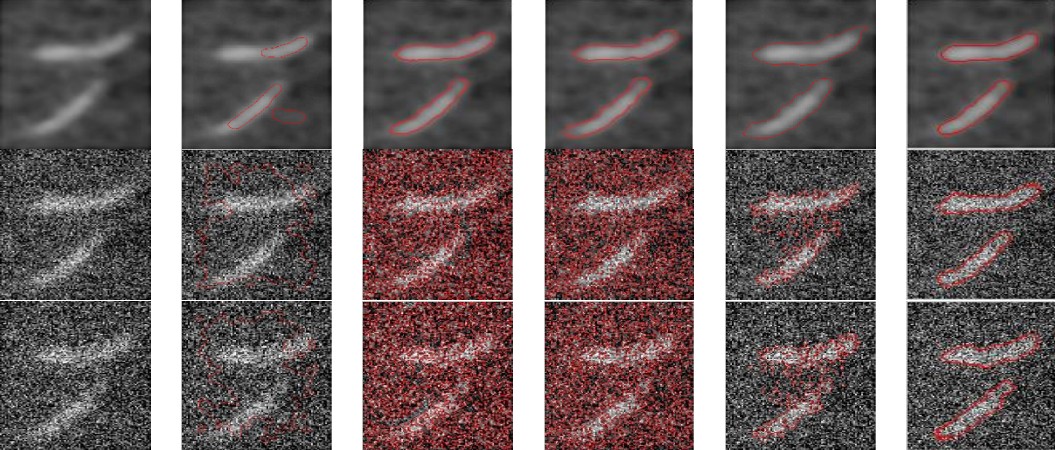

**Figure 5.** *Cont.*

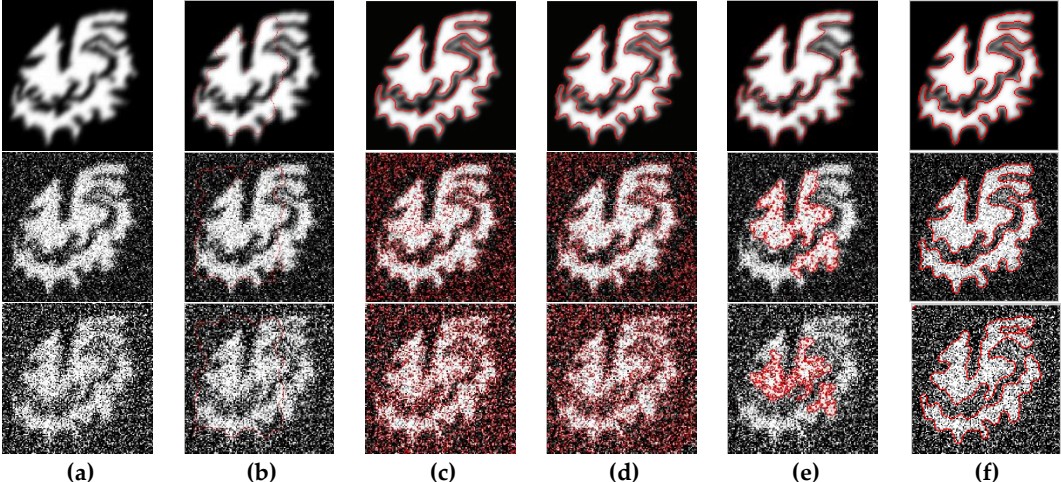

**Figure 5.** Comparisons of the five models in segmenting noisy images: (**a**) The original image; (**b**) The segmentation results of the DRLSE model; (**c**) The segmentation results of the LBF model; (**d**) The segmentation results of the LIF model; (**e**) The segmentation results of the LSACM model; (**f**) The segmentation results of the IRLS-IS model.

### 4.5. Segmentation of Medical Images

This part of our experiments is to verify the segmentation performance of medical images. First, we chose two simple medical images to demonstrate the segmentation performance of our IRLS-IS method. Figure 6 shows the original images and the segmentation results of the five compared methods, where the original simple medical images can be obtained from [42,75], while the compared models are still DRLSE, LBF, LIF, and LSACM. The compared four models cannot segment the first image, and the results are shown in Row 1 of Figure 6. The second image in the second column shows that most of the boundaries are obtained by the DRLSE model. However, the internal details are not recognized; the results are illustrated in Figure 6b. The object boundaries are not accurately extracted by the LBF, LIF, and LSACM algorithms, as depicted in Figure 6c,d,e. The IRLS-IS model initializes the level set function as a constant, and this gets rid of the dependency on the initialization position, so the proposed IRLS-IS method can process some simple medical images efficiently. The segmentation results demonstrate that our IRLS-IS model achieves satisfactory results for simple medical images.

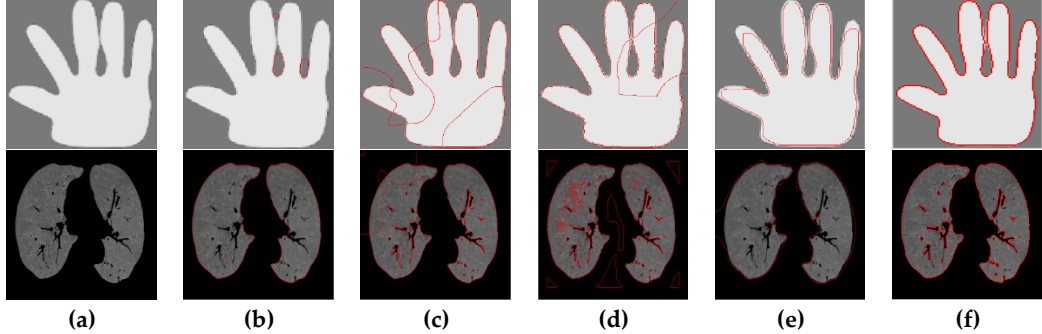

**Figure 6.** Comparisons of the five models in segmenting simple medical images: (**a**) The original image; (**b**) The segmentation results of the DRLSE model; (**c**) The segmentation results of the LBF model; (**d**) The segmentation results of the LIF model; (**e**) The segmentation results of the LSACM model; (**f**) The segmentation results of the IRLS-IS model.

Next, we continued testing our algorithms with two types of complex medical images. It is still a challenge to segment some complex medical images, because these medical images not only contain noise and intensity inhomogeneity but also have less obvious boundaries. To verify the efficiency

and feasibility of our proposed IRLS-IS method on complex medical images, three existing methods, namely, the LSACM algorithm [51], the cross entropy-based model (CEM) [46] and the local hybrid image fitting model (LHIF) [77], are selected to conduct further comparative experiments, and the results are presented in Figure 7, where the original image Figure 7a is from [34,77] and Figure 7c,d were previously obtained by Wang et al. [77].

　　Figure 7 presents experimental results of heart image segmentation with the four compared models. The image size is 152 × 128. As shown in Figure 7b, the LSACM cannot extract the true boundary of the heart image. Figure 7c illustrates that the CEM model has very poor performance in segmenting the heart image. The true boundaries are precisely obtained by Wang's LHIF model and our proposed IRLS-IS model, and the results are described in Figure 7d,e, respectively. There is a slight difference between the segmentation results of the LHIF model and the IRLS-IS model. Local leakage exists in the upper part of Figure 7d, but this is not the case in our IRLS-IS model. However, our model did not recognize the lower part of the right of Figure 7a, primarily because our algorithm has a large time step. Therefore, the experimental studies pertaining to the heart image state that the IRLS-IS algorithm efficiently eliminates the interference of intensity inhomogeneity and can achieve better segmentation performance on this medical image of a heart.

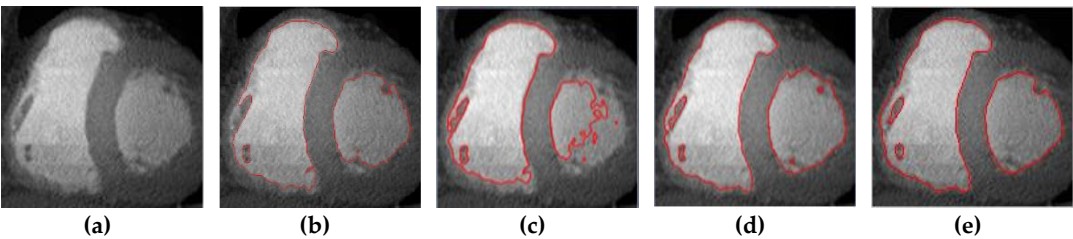

|  (a)  |  (b)  |  (c)  |  (d)  |  (e)  |

**Figure 7.** Comparisons of the four models in segmenting the heart image: (**a**) The original image; (**b**) The segmentation results of the LSACM model; (**c**) The segmentation results of the CEM model; (**d**) The segmentation results of the LHIF model; (**e**) The segmentation results of the IRLS-IS model.

　　When segmenting images with a cerebral infarction, there are two relatively sensitive problems with the traditional level set model [78,79]. One problem is that the boundaries between brain tissues are often not characterized by class changes; therefore, the boundary is prone to leakage. The other is that segmenting images of cerebral infarction is not a simple target and background segmentation problem. The medical images always incorporate various organizations and backgrounds. Thus, the traditional level set algorithms are not appropriate for brain images. Hence, the objective of the following portion of our experiments is to demonstrate the effectiveness of our IRLS-IS method on a complex cerebral infarction image. Our IRLS-IS model is compared with the DRLSE model [27] and the LSACM algorithm [51]. Figure 8 presents the results of the segmentation experiment, where the original cerebral infarction image can be downloaded at https://medpix.nlm.nih.gov/home (Picture Name: Posterior Reversible Leukoencephalopathy, PRES), and the image size is 369 × 351.

　　Figure 8b,c show that the DRLSE and LSACM algorithms cannot segment the cerebral infarction image. The segmentation results of our IRLS-IS method are demonstrated in Figure 8d. Since the IRLS-IS algorithm initializes the level set function to a constant value and breaks away from the dependence on the initialization position, our IRLS-IS method can analyze complex medical images very well.

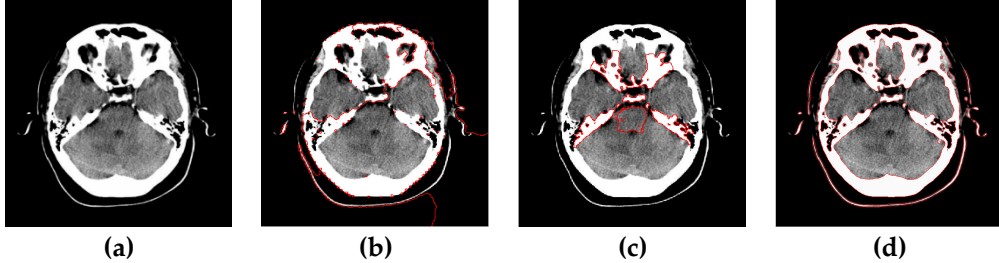

**Figure 8.** Comparisons of the three models in segmenting an image with a cerebral infarction: (**a**) The original image; (**b**) The segmentation results of the DRLSE model; (**c**) The segmentation results of the LSACM model; (**d**) The segmentation results of the IRLS-IS model.

*4.6. Comparative Evaluation*

In addition to the visual evaluation, the DICE coefficient (*Dice*) [80] and the Jaccard similarity index (*JSI*) [81] are often used as two important criteria for evaluating the segmentation accuracy of a target region. Following the experimental techniques designed in [34,75], the tested images were selected randomly from the BSDS500 database. Note that BSDS500 contains hundreds of natural images and their ground-truth segmentation maps generated by multiple individuals [82]. To enhance the coherency of our work with the above algorithms and compare it well with the CV model [48], the LBF model [49], and the LIF model [50], this next batch of experiments is conducted to further test our proposed IRLS-IS model on fifteen representative real-world color images, which are chosen from the Berkeley segmentation data set 500 (BSDS500).

*Dice* characterizes spatial overlaps between the segmented regions and the ground truth, and the formula of *Dice* [77] is expressed as

$$Dice(G, S) = \frac{2N(A \cap B)}{N(A) + N(B)}, \tag{17}$$

where $\cap$ represents an intersection operator, $N(\cdot)$ describes the number of pixels in the enclosed set, A represents the results of the IRLS-IS model to segment images, and B describes the ground truths. It is noted that the value of Dice is between 0 and 1. If the Dice is closer to 1, the segmentation results are more accurate [59]. Then, the Dice results of the four compared models are depicted in Table 1, where the best values are in bold font.

The *JSI* is the second statistical measure used for the quantitative evaluation in this paper, and the formula for the *JSI* [83] is

$$JSI(s_1, s_2) = \frac{|s_1 \cap s_2|}{|s_1 \cup s_2|}, \tag{18}$$

where $s_1$ describes the segmented volume and $s_2$ denotes the ground truth. The value of JSI is from 0 to 1. It is obvious that when the value of JSI is closer to 1, the more similar $s_1$ is to $s_2$.

Then, the precision of segmenting the Berkeley color images is measured by the *JSI* value, as shown in Table 1, where the bold font indicates the best result. As we can see from Table 1, the IRLS-IS method achieves the best values for the *Dice* and *JSI* on the thirteen-image data. On image ID: 102061, the *Dice* and *JSI* values of IRLS-IS are 0.01% and 0.02% less than those of CV, respectively, so they are almost the same. On image ID: 147091, the values of IRLS-IS are 0.2% and 0.3% less than those of CV, respectively, and then they only have subtle differences. Thus, the proposed IRLS-IS algorithm obtains the optimal values of *Dice* and *JSI* on the selected fifteen image data. In summary, these results demonstrate that our IRLS-IS method is indeed efficient and outperforms the three currently available approaches.

**Table 1.** The *Dice* and *JSI* values of the results of image segmentation on the fifteen Berkeley color images. *JSI*: Jaccard similarity index; CV: Chan Vese.

| Image ID | CV | | LBF | | LIF | | IRLS-IS | |
|---|---|---|---|---|---|---|---|---|
| | *Dice* | *JSI* | *Dice* | *JSI* | *Dice* | *JSI* | *Dice* | *JSI* |
| 8068 | 0.9780 | 0.9570 | 0.9555 | 0.9149 | 0.8673 | 0.7657 | **0.9792** | **0.9592** |
| 17067 | 0.9093 | 0.8336 | 0.8783 | 0.7830 | 0.8303 | 0.7099 | **0.9425** | **0.8912** |
| 28083 | 0.9469 | 0.8991 | 0.9236 | 0.8580 | 0.8062 | 0.6753 | **0.9556** | **0.9151** |
| 29030 | 0.9525 | 0.9093 | 0.9432 | 0.8925 | 0.8106 | 0.6815 | **0.9634** | **0.9293** |
| 33044 | 0.9108 | 0.8361 | 0.8700 | 0.7699 | 0.8253 | 0.7026 | **0.9434** | **0.8929** |
| 41004 | 0.9763 | 0.9537 | 0.9565 | 0.9166 | 0.8769 | 0.7808 | **0.9769** | **0.9548** |
| 41085 | 0.9209 | 0.8534 | 0.8818 | 0.7886 | 0.8446 | 0.7310 | **0.9229** | **0.8568** |
| 86016 | 0.9345 | 0.8770 | 0.7868 | 0.6485 | 0.7481 | 0.5976 | **0.9536** | **0.9113** |
| 102061 | **0.9634** | **0.9293** | 0.9442 | 0.8944 | 0.8516 | 0.7416 | 0.9633 | 0.9291 |
| 135069 | 0.9914 | 0.9829 | 0.9909 | 0.9819 | 0.8202 | 0.6951 | **0.9950** | **0.9900** |
| 143090 | 0.9575 | 0.9185 | 0.9517 | 0.9083 | 0.8633 | 0.7595 | **0.9715** | **0.9446** |
| 147091 | **0.9693** | **0.9404** | 0.9387 | 0.8844 | 0.8254 | 0.7027 | 0.9673 | 0.9367 |
| 207056 | 0.9677 | 0.9375 | 0.9305 | 0.8700 | 0.8183 | 0.6924 | **0.9874** | **0.9482** |
| 296059 | 0.9470 | 0.8994 | 0.9276 | 0.8650 | 0.8283 | 0.7069 | **0.9658** | **0.9339** |
| 317080 | 0.9591 | 0.9214 | 0.9288 | 0.8671 | 0.8665 | 0.7645 | **0.9596** | **0.9223** |

The abovementioned experimental results and analyses fully demonstrate the validity and stability of our IRLS-IS method. The new distance regularization term proposed in this paper not only avoids the problem of reinitialization but also initializes the level set function to a constant. It effectively solves the problem of selecting the initial size of the level set. In addition, the new distance regularization term is more robust than that proposed by Li and allows the partial differential equations to evolve with a large time step. Therefore, our IRLS-IS model successfully segments several noisy images and inhomogeneous images, efficiently handles some complex medical images, and reduces the iterations and the running time of the central processing unit (CPU) of computer.

*4.7. Discussion*

According to the abovementioned experimental results, our contributions to the IRLS-IS method can be summarized as follows.

(1) Compared with Li's DRLSE model in [18], our proposed IRLS-IS model initializes the level set function to a constant, which solves the problem of selecting the initial size of the level set and is insensitive to the initialization. The IRLS-IS model can eliminate the dependence on this initial contour position so that it can segment images effectively. However, the DRLSE model is overly reliant on the initialization location. Therefore, some numerical errors may occur during the evolution, and occasionally the desired segmentation results may not be derived [84–86]. Because only the local region information is used, the LBF method efficiently segments the inhomogeneous images. However, it is very easy to get caught up in a local minimum when this initial contour is inappropriate. Thus, the LBF method is sensitive to the initial contour of the images. In contrast to LBF, LIF conducts the convolution before the iteration, and thus, the running time of the computing can be reduced dramatically [87,88], but the initial contour of LBF and LIF is weak in robustness. The global information-based CEM model is the same as the CV model and cannot achieve the satisfied segmentation results on some inhomogeneous images.

(2) The LoG filter-based external energy term is presented. The external energy term not only reduces the influence of noise on image segmentation but also overcomes the interference caused by intensity inhomogeneity. Moreover, the internal energy term is introduced to regulate the smoothness of the zero level curves and to eliminate the low occurrence and isolated regions in the final segmentation results [23].

(3) The new distance regularization term corrects this deviation of the traditional level set function and the SDF to avoid the expensive computation of reinitialization. In addition, the new distance regularization term makes the IRLS-IS method evolve in a large step, and thus, the IRLS-IS method can

improve the efficiency and generate the results of image segmentation. Therefore, our IRLS-IS model has a lower computational complexity than Li's DRLSE model in [27].

## 5. Conclusion

In this work, we have introduced a new potential function and reconstructed a distance regularization term to compensate for the deviation between the traditional level set function and the SDF. Furthermore, the new distance regularization term can avoid this periodically reinitialized operation of the level set function. Our proposed distance regularization term is employed for the evolution of the partial differential equation, which guarantees the computational accuracy of the level set function. To efficiently validate the segmentation performance of the presented IRLS-IS algorithm, several experiments are conducted in this study. The results fully show that our proposed IRLS-IS method can effectively analyze noisy images and inhomogeneous images. Therefore, the IRLS-IS algorithm not only segments the specified target well but also reduces the boundary leakage, which improves the accuracy and robustness of the segmentation. However, taking account into the uncertainty of medical images, such as blurred boundaries and gray matter, and because the medical image itself has low contrast (such that the boundary between the tissue and lesions is blurred), the proposed method will not be suitable in all situations.

**Author Contributions:** L.S. and J.X. conceived the algorithm and designed the experiments; X.M. implemented the experiments; S.Z. analyzed the results; X.M. drafted the manuscript. All authors read and revised the manuscript.

**Funding:** This research was funded by the National Natural Science Foundation of China (Grants 61772176, 61402153, 61370169, and U1604156), the China Postdoctoral Science Foundation (Grant 2016M602247), the Plan for Scientific Innovation Talent of Henan Province (Grant 184100510003), the Key Project of Science and Technology Department of Henan Province (Grant 182102210362), the Young Scholar Program of Henan Province (Grant 2017GGJS041), the Key Scientific and Technological Project of Xinxiang City (Grant CXGG17002), the Natural Science Foundation of Henan Province (Grant 182300410130), and the Ph.D. Research Foundation of Henan Normal University (Grant qd15132).

**Conflicts of Interest:** The authors declare no conflict of interest.

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
