# Peer review of "An Image Segmentation Method Based on Improved Regularized Level Set Model"

_applsci, doi:10.3390/app8122393_

Round 1

Reviewer 1 Report

An Image Segmentation Method Based on an Improved Regularized Level Set Model

Image segmentation is a popular research area and a great deal has been published on the subject. This paper concentrates on one new approach to image segmentation and provides some examples of its use.

The English in the paper is understandable, but copy editing is required.

The introduction is comprehensive and provides good references to other work done in the area. But it's important to include some more general image segmentation background information to show where the techniques described in the paper fit into the wider subject area. For example, what's wrong with simple approaches of intensity thresholding? Are boundary-following approaches better? That sort of thing.

The second section (Related Knowledge - perhaps better expressed as "Other Related Work") is fine, but may benefit from a few more words of gentle introduction and scene-setting. The third section describing the proposed method is also fine (although I didn't quite have enough time to check all the maths...!).

But I have some concerns with section four onwards. These concerns generally relate to the complex task of justifying the use of this new technique to a highly complex area such as medical images.

Line 196 - it's important to discuss the issue of parameter selection in a practical context.

Line 201 - why these particular image segmentation approaches? The choice needs to be justified.

Line 206 and figure 2 - it's really hard to see the red pixels indicating the attempt at segmentation. They wouldn't be very visible in a printed version of the paper. It may be better to show the segmentation mask instead. And it's not completely clear what aspect of each image you are trying to identify. This is why it is important to explain the choice of segmentation parameters.

Line 216 - making that strong positive statement about the IRLS-IS approach isn't really justified - maybe the results can be said to be "promising", or something.

Line 228 - same considerations in figure 3 as in figure 2.

Section 4.4 - it's easy to do a quantitative (and statistically valid) study by adding varying levels of artificial noise to the images.

Section 4.5 - I'm a little worried by the simplicity of these "medical" images. I'd guess that other simpler segmentation algorithms can give better results with these images, and the statement on lines 272-3 can't be justified.

Line 287 - the heart segmentation needs some more information - what is the correct segmentation? Is it clinically useful?

Line 306-7 - "not a simple target and background segmentation problem" - that's exactly right! That's what I'm trying to say! What are you trying to segment in the cerebral infarction image?

Line 313 - which particular image on that web site?

Section 4.6 - 9 is a very small sample in order to obtain statistical significance. The statement on line 338 is therefore hard to justify.

Figures 8 and 9 shouldn't be graphs - they should be combined into a single table.

Lines 351-8 - tone down the statement of full verification of the validity of the approach - it's unjustified.

Lines 399-400 - yes, that's true. It would be good to expand on this statement a little.

To summarise, there's a great deal of useful, sensible work described here and the proposed approach to segmentation looks promising, but the analysis and evaluation you have presented needs attention, especially when applied to the field of medical imaging. Would it be appropriate to concentrate on simpler images and look at, for example, the effect on segmentation of the addition of various levels of noise?

Author Response

Dear,

We are very grateful to you for your valuable comments and suggestions. We have carefully revised the paper in accordance with these comments and suggestions. The added and modified parts are shown in red in the revised manuscript (and changes are marked). The main revisions are as follows.

Comment 1: The English in the paper is understandable, but copy editing is required.

Response: Thank you very much for your valuable suggestion.

We have corrected the mistakes in this revised manuscript. In addition, American Journal Experts helped us revise the entire manuscript carefully to correct English grammar, spelling, and sentence structure so that the goals and results of the study are clear to the reader.

Comment 2: The introduction is comprehensive and provides good references to other work done in the area. But it's important to include some more general image segmentation background information to show where the techniques described in the paper fit into the wider subject area. For example, what's wrong with simple approaches of intensity thresholding? Are boundary-following approaches better? That sort of thing.

Response: Thank you very much for your valuable suggestion.

We have added some more general image segmentation background information on where the techniques described in the paper fit into the wider subject area as follows:

On Pages 1-2: The recently developed methods of image segmentation can be roughly divided into two categories: traditional methods and soft computing methods [2]. Since the former methods are simple and easy to implement, they have been widely applied and provide exact solutions to potential and practical applications. In general, considering their form of operation, these methods can be grouped into different classes of clustering, thresholding, boundary tracking, region-based segmentation, and edge-based segmentation, among others [4]. In the last few years, many clustering-based models have been used in image segmentation. In particular, those algorithms based on K-means clustering, fuzzy c-means clustering (FCM) and spectral clustering are the most widely used in image segmentation problems [5]. Wang et al. [6] presented an adaptive segmentation method for crop disease images based on K-means clustering to improve the accuracy and stability of disease spot segmentation. As the FCM algorithm is sensitive to noise and selection to the initial cluster centers, some improvements to the FCM model have been investigated for both the estimation of the intensity inhomogeneity and segmentation of magnetic resonance image data [7]. Ramudu et al. [8] combined kernel FCM clustering with particle swarm optimization to develop a global region-based image segmentation method with a fast level set model. Hu et al. [9] proposed an adaptive kernel-based FCM clustering with spatial constraints model to automatically control the influence of the neighborhood pixels on the central pixel. However, its drawbacks include the slow convergence of the iteration and oversmoothing when suppressing noise [10]. He et al. [11] investigated an incremental spectral clustering method for stream image segmentation. It is known that for traditional spectral clustering, a scaling parameter needs to be fixed artificially, and obtaining its optimal value is very difficult in a Gaussian kernel function. Thus, handling the scaling parameter is a sensitive task. To solve this issue, Zelnik-Manor and Perona [12] constructed a self-tuning method for spectral clustering and image segmentation with a local scaling parameter. Goyal et al. [13] studied a fuzzy similarity measure-based spectral clustering method for noisy image segmentation, which improves the robustness to the influence of noise. However, most of these models share some common drawbacks, such as the cluster number needing to be given in advance and the sensitivity to selecting initial cluster centers. In addition, when low-level features are considered to make the representations in most algorithms, some crucial information may be lost. As a result, the segmentation performance may be degraded for some images to some extent. Over several years, the thresholding-based image segmentation method has drawn broad attention, and numerous thresholding techniques have been developed. Since they are simple and robust to noisy images, the threshold-based segmentation methods have been widely applied. The most representative method is the Otsu between-class variance. However, its computation grows exponentially when more threshold values are incorporated [14]. Tobias and Seara [15] presented image segmentation by histogram thresholding using fuzzy sets according to the similarity between gray levels, which is easy to implement and has a low computational burden. However, it does not consider the spatial contextual information in the threshold selection process. Singla and Patra [16] designed a fast context-sensitive threshold selection technique to solve image segmentation problems. Ananthi et al. [17] investigated L-interval-valued intuitionistic fuzzy sets and set the least entropy as the threshold to segment the image. However, the method may lead to increased computation time, especially when dealing with multithreshold image segmentation [18]. Gao et al. [19] introduced an enhanced artificial bee colony optimizer into multilevel threshold image segmentation. For the traditional interactive image segmentation, the robots only work for region-based methods, excluding the important class of approaches that rely on the boundary tracking paradigm. Then, Spina and Falcao [20] proposed robot users to simulate human user behavior when segmenting an image through the addition of anchor points close to the object’s boundary. Miranda et al. [21] developed an optimum user-steered boundary tracking approach for image segmentation, which simulates the behavior of water flowing through a riverbed. However, the inefficiency of these approaches lies in the fact that they cannot deal with real-life complex problems that are tolerant of partial truths, imprecision, uncertainty, and approximations [2]. Note that none of these methods are good at segmenting a variety of images, and they usually perform well in one class of images while performing poorly in another class of images. In recent years, the active contour model (ACM) [22] has become one of the most promising frameworks and effective and accurate methods for image segmentation. Currently, most existing ACMs can be categorized into two basic classes: edge-based models and region-based models, according to image features for segmentation.

Comment 3: The second section (Related Knowledge - perhaps better expressed as "Other Related Work") is fine, but may benefit from a few more words of gentle introduction and scene-setting. The third section describing the proposed method is also fine (although I didn't quite have enough time to check all the maths...!).

Response: Thank you very much for your valuable suggestion.

On Pages 4, and 5-7, we have made the suggested revision.

Comment 4: Line 196 - it's important to discuss the issue of parameter selection in a practical context.

Response: Thank you very much for your valuable suggestion.

We added experiments to discuss the issue of parameter selection in a practical context as follows:

On Pages 7 and 8: Some related parameters that are set in the process of image segmentation are illustrated in Figure 2; the original images can be found in [3, 42, 67].

On Page 8: Figure 2 shows the IRLS-IS method involves two parameters to be manually set. Figure 2(a) illustrates the original image (left) and the experimental results (right) by setting σ = 4, 3.2, 2.6 and 2 with ν = 0.15. Figure 2(b) shows the original image (left) and the experimental results (right) by setting ν = 0.4, 0.2, 0.15 and 0.1 with σ = 2.6. The other related parameters are set as Δt = 5, λ = 5 and μ = 0.01.

Comment 5: Line 201 - why these particular image segmentation approaches? The choice needs to be justified.

Response: Thank you very much for your valuable question.

The reason for selecting these particular image segmentation approaches is as follows:

On Page 8: The DRLSE model [27] can maintain the regularity of the level set function, particularly the desirable signed distance property in the vicinity of the zero level set. The LBF model [49], as a region-based ACM for image segmentation in a variational level set framework, effectively uses local image information to segment the intensity-inhomogeneous images. The LIF model [67] has less computational complexity than the LBF model and does not need reinitialization for image segmentation. The LSACM [51] includes two-phase level set models and yields closed-form solutions for the estimated parameters in the distribution. These representative ACM algorithms are four state-of-the-art level set methods published recently for image segmentation. They show improvements over the classical ACM and are specially selected based on the level set method for comparison experiments. The chosen parameters for the four models can be found in [27, 49, 51, 67]. The experimental results are demonstrated in Figure 3, and the original images shown in Figure 3(a) can be found in [52, 75]. Following the experimental techniques for image segmentation in [26-51] in which the boundaries of objects were labeled in red, we used the same techniques to mark the boundaries of the target regions in this paper.

Comment 6: Line 206 and figure 2 - it's really hard to see the red pixels indicating the attempt at segmentation. They wouldn't be very visible in a printed version of the paper. It may be better to show the segmentation mask instead. And it's not completely clear what aspect of each image you are trying to identify. This is why it is important to explain the choice of segmentation parameters.

Response: Thank you very much for your valuable suggestion.

  First, to the best of our knowledge, the segmentation mask is mainly used in fields of deep learning, neural networks, semantic segmentation, etc. Some additional interesting segmentation masks will be carried out in our future work to further study the semantic segmentation of images.

Second, the choice of segmentation parameters is as follows:

On Page 8: Figure 2 shows the IRLS-IS method involves two parameters to be manually set. Figure 2(a) illustrates the original image (left) and the experimental results (right) by setting σ = 4, 3.2, 2.6 and 2 with ν = 0.15. Figure 2(b) shows the original image (left) and the experimental results (right) by setting ν = 0.4, 0.2, 0.15 and 0.1 with σ = 2.6. The other related parameters are set as Δt = 5, λ = 5 and μ = 0.01.

Third, the aspect of each image we are trying to identify is described as follows:

On Page 8: It is well known that the objective of image segmentation is to segment the whole image domain into several distinct regions in line with the regional consistency. To the best of our knowledge, most existing image segmentation techniques [2] only segment the target regions of images, which are usually not identified by tags and have no special contents. Our proposed IRLS-IS method is different from the approaches for the semantic segmentation of an image [71-74], in which the task is to predict the pixel-level category labels and recognize the objects in the image and segment them.

Comment 7: Line 216 - making that strong positive statement about the IRLS-IS approach isn't really justified - maybe the results can be said to be "promising", or something.

Response: Thank you very much for your valuable suggestion.

We have modified this text in the revised manuscript.

Comment 8: Line 228 - same considerations in figure 3 as in figure 2.

Response: Thank you very much for your valuable suggestion.

Similar to Comment 6, we have provided an explanation in the text.

Comment 9: Section 4.4 - it's easy to do a quantitative (and statistically valid) study by adding varying levels of artificial noise to the images.

Response: Thank you very much for your valuable suggestion.

Some quantitative studies on varying the levels of artificial noise submitted to the images are added as follows:

  On Pages 9 and 10: Figure 5 shows the original images with different intensity noise and the comparison results of the five segmentation methods, where the original images without noise in Figure 5(a) are coming from [76, 77]. In Figure 5, Row 1 and Row 6 are the original images and the segmentation results, respectively. Row 2 and Row 3 present noisy images (zero mean, variance 0.04 and 0.06) and the segmentation results, respectively. Row 4 and Row 5 also present noisy images (zero mean, variance 0.1 and 0.3) and the segmentation results. Figure 5(b) and 5(e) show that the DRLSE and LSACM algorithms cannot eliminate the interference of noise and fail to segment the images. It can be observed from Figure 5(c) and 5(d) that the LBF and LIF models could analyze the images without Gaussian noise well, but for the Gaussian noisy images, they perform segmentation poorly. As shown in Figure 5(f), the object boundaries are accurately extracted by our IRLS-IS method. The improvements in the results are mainly due to the external energy term, which can successfully address the noise problem. It can be clearly seen that the IRLS-IS model effectively eliminates the noise interference and completes the segmentation of the noisy image.

Comment 10: Section 4.5 - I'm a little worried by the simplicity of these "medical" images. I'd guess that other simpler segmentation algorithms can give better results with these images, and the statement on lines 272-3 can't be justified.

Response: Thank you very much for your valuable suggestion.

Other, simpler segmentation algorithms may achieve better results with these images, but I have not found them after searching for several days, except for the ACM algorithm.

Comment 11: Line 287 - the heart segmentation needs some more information - what is the correct segmentation? Is it clinically useful?

Response: Thank you very much for your valuable question.

First, on Page 8: It is well known that the objective of image segmentation is to segment the whole image domain into several distinct regions in line with the regional consistency. To the best of our knowledge, most existing image segmentation techniques [2] only segment the target regions of images, which are usually not identified by tags and have no special contents. Our proposed IRLS-IS method is different from the approaches for the semantic segmentation of an image [71-74], in which the task is to predict the pixel-level category labels and recognize the objects in the image and segment them.

Second, in the field of image segmentation, to the best of our knowledge, the ground-truth segmentation is generally marked by experts manually. We only use this image to prove the validity of our proposed algorithm compared with the other related algorithms. The ground-truth segmentation of the test heart image has not been found yet. In addition, some work about the clinical application of heart segmentation will be carried out in our future work.

Comment 12: Line 306-7 - "not a simple target and background segmentation problem" - that's exactly right! That's what I'm trying to say! What are you trying to segment in the cerebral infarction image?

Response: Thank you very much for your valuable question.

Similar to Comment 11, we have provided an explanation in the text.

Comment 13: Line 313 - which particular image on that web site?

Response: Thank you very much for your valuable suggestion.

On Page 12: Figure 8 presents the results of the segmentation experiment, where the original cerebral infarction image can be downloaded at https://medpix.nlm.nih.gov/ home (Picture Name: Posterior Reversible Leukoencephalopathy, PRES), and the image size is 369×351.

Comment 14: Section 4.6 - 9 is a very small sample in order to obtain statistical significance. The statement on line 338 is therefore hard to justify.

Response: Thank you very much for your valuable suggestion.

The images used for testing have been extended, and the experimental results are described as follows:

On Pages 12 and 13: To enhance the coherency of our work with the above algorithms and compare it well with the CV model [48], the LBF model [49] and the LIF model [67], this next batch of experiments is conducted to further test our proposed IRLS-IS model on fifteen representative real-world color images, which are chosen from the Berkeley segmentation data set 500 (BSDS500).

On Page 13: As we can see from Table 1, the IRLS-IS method achieves the best values for the Dice and JSI on the thirteen-image data. On image ID: 102061, the Dice and JSI values of IRLS-IS are 0.01% and 0.02% less than those of CV, respectively, so they are almost the same. On image ID: 147091, the values of IRLS-IS are 0.2% and 0.3% less than those of CV, respectively, and then they only have subtle differences. Thus, the proposed IRLS-IS algorithm obtains the optimal values of Dice and JSI on the selected fifteen image data. In summary, these results demonstrate that our IRLS-IS method is indeed efficient and outperforms the three currently available approaches.

Comment 15: Figures 8 and 9 shouldn't be graphs - they should be combined into a single table.

Response: Thank you very much for your valuable suggestion.

On Pages 13 and 14, we have combined Figures 8 and 9 into a single table in this revision, as shown in Table 1.

Comment 16: Lines 351-8 - tone down the statement of full verification of the validity of the approach - it's unjustified.

Response: Thank you very much for your valuable suggestion.

  We have modified the text in this revision.

Comment 17: Lines 399-400 - yes, that's true. It would be good to expand on this statement a little.

Response: Thank you very much for your valuable suggestion.

 We have expanded this statement as follows:

On Page 14: However, taking account into the uncertainty of medical images, such as blurred boundaries and gray matter, and because the medical image itself has low contrast (such that the boundary between the tissue and lesions is blurred), the proposed method will not be suitable in all situations.

Comment 18: To summarise, there's a great deal of useful, sensible work described here and the proposed approach to segmentation looks promising, but the analysis and evaluation you have presented needs attention, especially when applied to the field of medical imaging. Would it be appropriate to concentrate on simpler images and look at, for example, the effect on segmentation of the addition of various levels of noise?

Response: Thank you very much for your valuable suggestion.

Similar to Comment 9, we have provided an explanation in the text.

Thank you once again for your constructive and valuable comments.

Best wishes,

Prof. Lin Sun, Ph.D.,

College of Computer and Information Engineering, Henan Normal University

Email: linsunok@gmail.com

Reviewer 2 Report

The work is well presented, even if there are some decisions of the authors (in the design phase as well as in the experimental trials) that should be justified and addressed in order to properly evaluate this contribution.

Major issues:

Comment 1: The literature proposes very many approaches for the reinitialization of LSFs that exploit a regularization term. How did the authors arrive at the formulation of the proposed regularization term in the equation (6) ?

Comment 2: In the experimental results, the choice to present both a visual comparison and then a quantification through the Dice and Jaccard indices is good. The quantitative evaluation is done only on 9 images of the BSDS500 dataset (rows 326-328). Considering that in all these 9 images IRLS-IS obtains better results than the other approaches, it is natural to ask which criterion was used for the selection of the 9 images and for these 9 images? In order to have a more robust statistic it would be appropriate to extend the number of images used for testing.

Minor issues:

 - an overall English revision is needed;

Author Response

Dear,

We are very grateful to you for your valuable comments and suggestions. We have carefully revised the paper in accordance with these comments and suggestions. The added and modified parts are shown in red in the revised manuscript (and changes are marked). The main revisions are as follows.

Comment 1: The literature proposes very many approaches for the reinitialization of LSFs that exploit a regularization term. How did the authors arrive at the formulation of the proposed regularization term in the equation (6) ?

Response: Thank you very much for your valuable questions.

The process of arriving at the formulation of the proposed regularization term in equation (6) is as follows:

On Page 5: From Eqs. (4) and (6),                  can be obtained as the diffusion ratio of the level set evolution.

On Page 5: It is known that for the level set model, a potential function must have a point with a minimum on s = 1, and the objective of constructing a new potential function is to keep the SDF  in the vicinity of the zero level set. Then, owing to a new potential function, the regularized level set function can remain an SDF throughout the evolution process. Thus, a new potential function is constructed as

(7)

Due to the existing proportional relationship between the diffusion ratio and the potential function, the diffusion ratio can be defined as

(8)

Based on the energy penalty term Eq. (7) and the defined potential function Eq. (8), a new distance regularization term is defined as

(9)

The new distance regularization term corrects the deviation of the level set function and the SDF. Meanwhile, this term ensures that the level set evolves without requiring periodic initialization and can avoid a sharp or flat shape. Therefore, the reinitialization step can be avoided during the level set evolution.

Comment 2: In the experimental results, the choice to present both a visual comparison and then a quantification through the Dice and Jaccard indices is good. The quantitative evaluation is done only on 9 images of the BSDS500 dataset (rows 326-328). Considering that in all these 9 images IRLS-IS obtains better results than the other approaches, it is natural to ask which criterion was used for the selection of the 9 images and for these 9 images? In order to have a more robust statistic it would be appropriate to extend the number of images used for testing.

Response: Thank you very much for your valuable suggestion.

First, the criterion used for the selection of the 9 images and for these 9 images is described as follows:

On Pages 12 and 13: Following the experimental techniques designed in [34, 76], the tested images were selected randomly from the BSDS500 database. Note that BSDS500 contains hundreds of natural images and their ground-truth segmentation maps generated by multiple individuals [83]. To enhance the coherency of our work with the above algorithms and compare it well with the CV model [48], the LBF model [49] and the LIF model [67], this next batch of experiments is conducted to further test our proposed IRLS-IS model on fifteen representative real-world color images, which are chosen from the Berkeley segmentation data set 500 (BSDS500).

Second, the images used for testing have been extended as follows:

On Pages 12 and 13: To enhance the coherency of our work with the above algorithms and compare it well with the CV model [48], the LBF model [49] and the LIF model [67], this next batch of experiments is conducted to further test our proposed IRLS-IS model on fifteen representative real-world color images, which are chosen from the Berkeley segmentation data set 500 (BSDS500).

On Page 13: As we can see from Table 1, the IRLS-IS method achieves the best values for the Dice and JSI on the thirteen-image data. On image ID: 102061, the Dice and JSI values of IRLS-IS are 0.01% and 0.02% less than those of CV, respectively, so they are almost the same. On image ID: 147091, the values of IRLS-IS are 0.2% and 0.3% less than those of CV, respectively, and then they only have subtle differences. Thus, the proposed IRLS-IS algorithm obtains the optimal values of Dice and JSI on the selected fifteen image data. In summary, these results demonstrate that our IRLS-IS method is indeed efficient and outperforms the three currently available approaches.

Comment 3: An overall English revision is needed

Response: Thank you very much for your valuable suggestion.

We have corrected the mistakes in this revised manuscript. In addition, American Journal Experts helped us revise the entire manuscript carefully to correct English grammar, spelling, and sentence structure so that the goals and results of the study are clear to the reader.

Thank you once again for your constructive and valuable comments.

Best wishes,

Prof. Lin Sun, Ph.D.,

College of Computer and Information Engineering, Henan Normal University

Email: linsunok@gmail.com

Round 2

Reviewer 1 Report

Many thanks for carrying out extensive changes to the paper. It is much improved. I like the new introduction - it is readable and comprehensive and is a useful document on its own. The main body of the paper is also much improved and is much easier to read.

Reviewer 2 Report

Thanks to have properly addressed all the detected issues.